# Governing Noncommunicable Diseases Through Political Rationality and Technologies of Government: A Discourse Analysis

**DOI:** 10.3390/ijerph17124413

**Published:** 2020-06-19

**Authors:** Joshua S. Yang, Hadii M. Mamudu, Timothy K. Mackey

**Affiliations:** 1Department of Public Health, California State University, Fullerton, KHS 161 A, 800 N. State College Blvd, Fullerton, CA 92831, USA; 2Department of Health Services Management and Policy, College of Public Health, East Tennessee State University, Johnson City, TN 37614, USA; mamudu@mail.etsu.edu; 3Department of Anesthesiology and Division of Global Public Health, San Diego School of Medicine, University of California, San Diego, CA 92037, USA; tmackey@ucsd.edu; 4Global Health Policy Institute, San Diego, CA 92130, USA

**Keywords:** noncommunicable diseases, global governance, governmentality, discourse analysis

## Abstract

In the last two decades, global action to address noncommunicable diseases (NCDs) has accelerated, but policy adoption and implementation at the national level has been inadequate. This analysis examines the role of rationalities of governing, or *governmentality*, in national-level adoption of global recommendations. Critical discourse analysis was conducted using 49 formal institutional and organizational documents obtained through snowball sampling methodology. Text were coded using a framework of five forms of governmentality and analyzed to describe the order of discourse which has emerged within the global NCD policy domain. The dominant political rationality used to frame NCDs is rooted in risk governmentality. Recommendations for tobacco control and prevention of harmful alcohol use rely on a governmentality of police mixed with discipline. The promotion of physical activity relies heavily on disciplinary governmentality, and the prevention of unhealthy diet mixed disciplinary measures, discipline, and neoliberal governmentalities. To translate global NCD prevention and control strategies to national action, acceptability for the political rationalities embodied in policy options must be nurtured as new norms, procedures, and institutions appropriate to the political rationalities of specific interventions are developed.

## 1. Introduction

In the last 20 years, non-communicable diseases (NCDs) have emerged as a priority policy and governance area in global health. The World Health Report 1997 [1] provided the impetus for the World Health Assembly (WHA) to issue a resolution to develop a global strategy for the prevention and control of NCDs as part of health sector reform [2]. In 2000, the World Health Organization (WHO) Global Strategy for the Prevention and Control of NCDs was adopted by the WHA, representing the first effort to find a governance solution to the NCDs problem and to provide recommended actions to Member States and other international organizations [3]. Since that time, an array of resolutions, political declarations, global strategies, action plans, and progress reports have been published to hasten progress on shared governance for NCDs and related risk factors. Key achievements include convening the United Nations (UN) High Level Meeting (HLM) on NCDs and its resulting political declaration in 2011 [4,5]; defining nine global targets to reduce NCD burden [6]; agreeing to a global coordinating mechanism for NCDs [6]; establishing a WHO High Level Commission on NCDS [7]; and adopting global action plans for NCD prevention and control in 2008 and 2013 [8,9]. Numerous policy and technical documents since the mobilization of a foundational governance infrastructure have endeavored to sustain earlier momentum and commitments to address NCDs (see Table 1).

Despite increased emphasis on NCDs and the existence of known evidence-based interventions to address their increasing disease burden, policy adoption and implementation at the national level has been inadequate, with the majority of countries not on track to meeting the nine global NCD targets. These internationally agreed-upon targets include a 25% reduction in risk of premature mortality from NCDs, at least a 10% relative reduction in the harmful use of alcohol, 10% relative reduction in the prevalence of insufficient physical activity, 30% relative reduction in mean population intake of salt/sodium, and 30% reduction in the prevalence of current tobacco use in persons older than age 15 years [10,11,12]. Some have pointed to limited funds, weak health systems, poor framing and articulation of NCDs to the public, and the sheer complexity of an NCD-specific challenge as obstacles to progress [13]. What has received less attention is whether evidence-based interventions within the scientific literature and recommended by global institutions are acceptable to national governments and can be effectively implemented by Member States. Compatibility between the particular political ethos and governmental machinery of a Member State with the underlying governmental rationality inherent in an intervention may lead to greater uptake and implementation of recommended NCD prevention and control policies. In contrast, discordance between state political ethos and political rationality of interventions may lead to less uptake. 

The menu of policy options offered by global institutions are prioritized based on calculable, empirically derived criteria, and in particular their cost-effectiveness. They are treated as technical solutions to NCDs, but whether interventions are politically viable or can be effectively implemented given the institutions, laws, or national governance dynamics within a country is not considered as part of policy recommendations. Seen in this light, advocacy for NCD prevention and control policies is not only asking governments to accede to the empirical basis of an intervention but requesting that governments accept particular types of governing implicit in policy recommendations. In fact, the WHO High Level Commission on NCDs has pointed to the political domain as the primary impediment to greater uptake of recommended interventions [7]. A first step in overcoming political barriers is to make explicit the assumed governmental rationality inherent in recommended policy interventions. Once these rationalities are made clear, efforts to open political and governmental systems to NCD prevention and control policies can better specify needed political and governmental changes and allay concerns within entrenched structures that may accompany expanding or evolving governmental rationalities. The purpose of this analysis is to describe and distinguish the various political rationalities, or governmentalities, implicit in NCD prevention and control policy recommendations. We utilize the concept of governmentality to make explicit the implied forms of governing in global recommendations for action on NCDs and describe how multiple approaches to governing are interwoven into the broader program of NCD prevention and control. Therefore, this paper is consistent with the idea that addressing the persistent increase in NCD-related disease burden requires looking beyond individually driven factors and bringing structural issues into the dialogue and practice [5]. 

### 1.1. Governmentality

Governmentality is a concept linked most closely with Michel Foucault [14] and refers to the examination of how diverse forms of knowledge and methods are packaged into rationalities of governing. Within the health literature, Foucault’s work on governmentality has mainly been applied using the biopower concept in different areas of public health practice [15,16,17,18,19]. While numerous efforts have documented NCD policy processes at the national level [20,21,22,23], no studies have applied the concept of governmentality as a political rationality to NCDs policy. Foucault uses the term government broadly, viewing it as deliberate and rational action intended to shape human conduct through the desires, interests and beliefs of actors for defined ends; it is undertaken by any number of authorities and agencies and utilizes different forms of knowledge and a variety of techniques [24]. Foucault has traced government as a specific form of power that emerged in the sixteenth century. He describes the art of government, or governmentality, as “the ensemble formed by the institutions, procedures, analyses and reflections, the calculations and tactics that allow the exercise of the very specific albeit complex form of power, which has as its target population…” [14]. Historically, government has not always referred to the state, but included the government of one’s self, souls and lives, children, and families. Only since the eighteenth century has the governmentalization of the state been the dominant form of rule. Over time, the problems of government have been addressed through different rationalities. Governmental rationalities may predominate in different historical periods, but they do not exist exclusive of each other in any given period or within a particular jurisdiction. 

Dean [24] interprets governmentality as how we think about or rationalize governing. Implicit in Dean’s interpretation is systematic reasoning and calculation which draws upon expertise and formal bodies of knowledge to respond to problems. There are two dimensions of governmentality: political rationality and technologies of government [25]. *Political rationalities* are regularities discernable in political discourse, idealized representations of reality, formulated and justified to analyze a particular reality. Political rationalities have a moral form concerned with ideals and principles; an epistemological character that specifies the nature of objects governed and a distinctive “intellectual machinery or apparatus for rendering reality thinkable in such a way that it is amenable to political deliberations” [26]. *Technologies of government* are the methods of government action. They are not ideal types, but the specific packaging of governmental tools used to regulate individual, group, and organizational action. These technologies include particular domains (e.g., legal, administrative, financial, judgmental), methods (e.g., calculations, evaluations, examinations), and devices (surveys, charts, training systems) [25]. Specific forms of governmentality include discipline, pastoral power, liberalism, neoliberalism and the risk society. 

### 1.2. Discipline

This particular form of government emerged in 17th century Europe as a specific form of power exercised directly over and through individuals, the body and its capacities, and collectives, with the goal of regulating and ordering populations within territories in order to meet the goals of government. Discipline as a political rationality is an instrumental form of control, utilizing human capacities and energies for specific ends [27]. Discipline is not only a punitive, constraining force over whom it is exercised, it is also productive, helping to enhance individual capacities [28]. Individuals and collectives are viewed “calculable and manipulable sets of forces” [27], which are brought in line with broader goals through developing specific skills, attributes, and self-control; promoting the ability to work in unison; and shaping of character. Discipline depends on specialized knowledge by which individuals could be molded and manipulated into an optimal form. Expertise, especially in the behavioral and social sciences, are an essential companion to disciplinary power. Surveillance, regimentation, and classification are techniques of discipline used to bend individuals and collectives to instrumental goals. Educational campaigns to motivate individual behavior change are examples of a disciplinary governmentality. 

### 1.3. Pastoral Power

In contrast to discipline, pastoral power is based on the metaphor of a shepherd caring for her flock. The goal of pastoral power is ensuring the welfare of subjects through comprehensive regulation of behavior [27]. The shepherd is a superior being, without whom the flock would not exist, and who may require the flock to act without the need for consent. Pastoral power is best represented by the concept of police and expressed through the intellectual development of cameralism in continental Europe during the seventeenth and eighteenth centuries. Police in this context does not refer to contemporary officers designed to prevent and investigate crime. The science of police was concerned with the development of an administrative state with twin goals of ensuring the thriving, prosperity, and wellbeing of individuals and the expansion of state wealth and power. The method of police was a comprehensive and totalizing regulation of public and private life through rationally determined administrative rules. Police included a distinctly moral dimension of ensuring the general morality of populations and to preserve public decency [24], which were essential to achieving the prosperity of subjects and, more importantly, the power and prosperity of the state. Subjects’ responsibility was to respect and obey the dictates of the administrative state. Foucault [14] suggests that the model of pastoral power was an important precursor to the modern welfare state. Pastoral power would include tax measures on unhealthy products such as cigarettes or sugar-sweetened beverages. 

### 1.4. Liberalism

In response to an expansive state embodied by the science of police, liberalism emerged with skepticism toward the state and how well it could know—and thus regulate—the reality to be governed [29]. The early liberal challenge to the state occurred in the domain of the economy. Reflecting the natural philosophy of the Enlightenment period, the economy was assumed to follow natural laws with its own dynamics and self-regulatory capacity. State intervention in the economy would therefore only serve to distort the natural functioning of the economy. Laissez-faire was a doctrine of government non-interference and a justification for market freedom based on the idea of efficiency and that more benefit would accrue to the state through governing less. Liberal rationality extended this political economy to institutions of the state and of society [24]. The logic of liberalism requires state intervention to ensure the freedom of individuals but prevents it from interfering with the natural laws of economy and society. Thus, instead of the state determining and assuring the welfare of individuals and the population, liberalism places determination of wellbeing within the control and direction of the individual. The goals of government can thus be best realized through the free choices of individuals [27]. The liberal technologies of government focus on institutions that aim to create individuals who do not need to be governed because they are able to govern and care for themselves. Hindess [27] suggests that the success of liberalism in contemporary Western societies is that “the vast majority of those inhabitants have already been trained in the dispositions and value of responsible autonomy”. A liberal governmentality would argue for no, or minimal, government intervention to address NCDs. 

### 1.5. Neoliberalism

Unlike liberalism, neoliberalism is concerned not with limiting government intervention in the economy and society but how to expand the competitive market model into a generalized political rationality. The neoliberal role for government is not to exclude it from the economy and civil society, but to have it actively create the political, legal, and institutional environments within which market rationalities can be used as a form of government. Individuals are constructed as entrepreneurial, competitive, economic-rational actors. Foucault suggests that the neoliberal project is an “extension of economic analysis into a previously unexplored domain” [30] and uses the model of the market economy to “decipher non-market relationships and phenomena which are not strictly and specifically economic but what we call social phenomenon” [30] The market, redefined in neoliberalism as a site of competition, is the principle through which social relationships and individual behavior is analyzed, assessed, and understood: conduct becomes entrepreneurship, an advancement of the self. The enterprise nature is utilized to maximize quality of life through choice as determined by individual values and meaning [25]. The individual as entrepreneur is responsible for themselves and fulfills social obligations through pursuit of self-fulfillment within various sub-communities of life including family, work, school, and neighborhoods. The regulation of conduct therefore originates from within an individual who governs themself to maximize their own happiness and fulfillment. Individual choice is valorized, and market rationality driven by individual choice are the ideal form of governing. Neoliberal governmentality is exemplified by efforts to change the choice environment, such as for food, to make it easier for people to choose healthier options.

### 1.6. Risk

A final form of governmentality operates through the concept of risk, a form of calculative rationality, or any number of efforts to represent uncertainty into a calculable, and thus governable, form [24]. The diverse elements used to manage risk attempt to minimize potential harms in the present to maximize the potential of the future. Thus, the risk analysis tools used to develop specific forms of knowledge through statistics, sociology, epidemiology, management and accounting ultimately replace the individual with a set of risk factors used to govern populations. There are two dominant treatments of risk rationality, one which focuses on a sociological account of risk society associated with Ulrich Beck and a second related to the analytics of government [24]. Beck [31] defines risk as “a systematic way of dealing with hazards and insecurities induced and introduced by modernity itself.” Through industrialization and modernization, we have transitioned from societies of scarcity—where the goal was to meet material need—to risk societies which are defined by the need to manage the risks which have accompanied the production of wealth. Modernization has produced great wealth, but the sources of that wealth are also producing hazardous side effects, the results of which are increasingly apparent. Whereas wealth societies are concerned with wealth distribution, risk societies are concerned with the distribution of risk in society. Beck believes that as industrial society begins to see itself as a risk society, it will endure a process of questioning and reconsidering dominant social structures. As an approach to government, the calculative rationality of risk moves it away from notions of dangerousness into the domain of chance, probability, and randomness. Because risk can be calculated, it can thus be governed. In an effort to minimize risk, governments engage in a wide range of methods and modes to act on populations—a *dispositif* of risk [32]—ranging from the coercive to promoting a new prudentialism, which require individuals, families, and communities to take responsibility for minimizing their risk. Risk governmentality focuses on characterizing unhealthfulness as a calculable and controllable hazard.

Political rationalities and technologies of government are constituted to serve specific governmentalities, and how easily or how quickly they can be reconstructed, adapted, or otherwise repurposed to meet the needs of other governmentalities is unclear. Implementing a global set of NCD policy actions for Member States who operate under a diverse array of governmentalities is a political challenge that extends beyond technical prioritization of prevention and control actions. To the accelerate implementation of NCD prevention and control actions, we must develop a clearer understanding of the implicit governmentalities that Member States are being asked to adopt, along with specific prevention and control actions. Official documentation and proclamations of international institutions are guideposts from which the governmentalities of global NCD discourse can be ascertained. 

## 2. Materials and Methods 

Critical discourse analysis (CDA) is the analytical method used to examine the positioning of governmentalities in the construction of the global NCD prevention and control policy domain. CDA is “discourse analytical research that primarily studies the way social-power abuse and inequality are enacted, reproduced, legitimated, and resisted by text and talk in the social and political context” [33]. CDA includes a variety of approaches and methods to describe how language is used to exert power; Fairclough’s [34] approach is adopted in this analysis for its explicit treatment of discourse as ideological. He builds on Foucault’s notion of discourse as a domain of statements or a group of statements, specifically text, [35] and focuses on discourse as an effort to create normative views of reality. Discourse is the vehicle for competing ideologies to undergo hegemonic struggle to shape power relations. Orders of discourse—the whole of the discursive practices which emerge within an institution or domain and how they relate to each other—emerge as multiple and conflicting ideologies are continually rearticulated. In the present analysis, governmentalities are expressed in text within documents to specify normative roles of government. The combination of these statements is taken as competing political ideologies and constitutes the order of discourse for the NCD policy domain. The resulting NCD order of discourse has implications for how Member States select, construct, implement, and manage policies to reduce NCD burden and which actors exert and gain more power in doing so. Official documents produced from global NCD policy fora such as the UN constitute a unique corpus of textual material through which to examine the articulation and arranging of political rationalities for action on NCDs.

Snowball sampling was utilized to obtain documents selected for analysis up to 2018 when the Third HLM on NCDs was held. The collection of documents has been part of an ongoing research agenda in global NCD policy development since 2011. Beginning with WHO and UN webpages related to NCDs and the 2011 UN HLM, publicly available documents were downloaded, reviewed, and analyzed to assess the degree to which the five forms of governmentality described above served as the basis for framing NCDs and/or proposed policy actions. As documents were reviewed, additional documents referenced during document review were obtained and included in the document retrieval process. A total of 224 documents were identified. Inclusion criteria for the current analysis were: (1) address NCDs as a group of conditions, or one of the main physical NCD conditions as defined by WHO (i.e., diabetes, cancer, cardiovascular disease, pulmonary disease), or a major risk factor for NCDs as defined by WHO (i.e., tobacco use, physical inactivity, poor diet, unhealthy use of alcohol); (2) originate from a representative global forum of nation states (e.g., WHO, UN); and (3) assess actions of or provide recommendations to nation-states. In total, 49 documents were included in this textual analysis. 

Included documents originated primarily from WHO, WHA, and the UN. Though the documents come from a limited set of organizations, these organizations are made up of Member States who provide input and vote on approving documents. The order of discourse within the documents thus represents a deliberated, consensus position on NCDs. Text analysis utilized a deductive coding scheme based on the five forms of governmentality discussed earlier. No emergent forms of governmentality were considered in the analysis. Documents were imported into and coded using Atlas.ti (version 8), a qualitative data analysis software. Documents were reviewed to assess the overall purpose of the document and the nature of the document. Data were coded by discursive types by the first author (J.S.Y.). Each political rationality was treated as its own discursive type, and data were coded based on the governmentality expressed within a textual segment. Upon completion of coding, data were analyzed to assess interdiscursivity and intertextual chains and synthesized to describe the order of discourse which has emerged within the global NCD policy domain. The second author (H.M.M.) reviewed the textual data, coding scheme, and resulting thematic patterns for agreement in interpretation of data. Differences in interpretation were resolved through discussion that resulted in consensus decisions. 

## 3. Results

The retrieved documents fall into three categories: political documents, technical descriptive documents, and technical prescriptive documents (see Table 1). While there is considerable overlap in the content of the documents, each category of documents had a particular purpose. Political documents include resolutions and political declarations from the UN General Assembly, WHA, and other meetings of Member States. These documents, in response to the burden of NCDs, were statements of commitments from Member States to address NCDs or their risk factors. The most significant of these documents was the UN General Assembly Political Declaration on the Prevention and Control of NCDs [4], but also includes precursor documents, such as WHA Resolutions 51.18 [2] and 53.17 [3] and the Moscow Declaration [36], and follow-up documents, such as the UN Outcome Document of the High-Level Meeting of the General Assembly on the Review of the Progress Achieved in the Prevention and Control of NCDs [37]. Technical descriptive documents were produced exclusively by WHO and focused on presenting the burden of NCDs and their distribution, the distribution of NCD risk factors, and/or establishment of NCD policies and programs across geographical regions and populations. Key documents include The World Health Report [1,38], WHO Global Status Report on NCDs [11,39], and WHO NCD Country Profiles [40,41]. Technical prescriptive documents, mostly produced by WHO, provided specific recommendations to Member States on actions to be taken to prevent and control NCDs. The central documents were the Global NCD Strategy [42] and Global NCD Action Plans of 2008 and 2013 [8,9], but also the recommendations developed in reports focused on specific risk factors including tobacco use, diet and physical inactivity, and harmful alcohol use (see Table 1). 

### 3.1. Risk and the Framing of the NCDs “Epidemic”

The central political rationality on which the global NCD prevention and control movement is based is risk governmentality. The relationship between NCDs and risk is two-fold. On the one hand, there are very well defined, calculable behavioral risk factors that can cause and worsen the progression of NCDs. On the other, NCDs are characterized as a present risk to human health and social and economic development of individuals and nations, with projections for continued and increasing harm in the future, including in the context of individual and population health, economic development, and societal issues such as equity and gender. Risk rationality is presented in an early political document, which recognized:

“The enormous human suffering caused by NCDs such as cardiovascular diseases, cancer, diabetes and chronic respiratory diseases, and the threat they pose to the economies of many Member States, leading to increasing health inequalities between countries and populations… that the conditions in which people live and their lifestyles influence their health and quality of life, and that the most prominent NCDs are linked to common risk factors, namely, tobacco use, alcohol abuse, unhealthy diet, physical inactivity, environmental carcinogens and being aware that these risk factors have economic, social, gender, political, behavioural and environmental determinants” [3].

Defining NCD risk in terms of disease, health, and economic development relies on epidemiological and econometric tools to characterize risk as well as epidemiological and economic (specifically cost-effectiveness) measures for the surveillance and monitoring of NCD actions. The most detailed and clearly articulated presentation of risk rationality is The World Health Report 2002, *Reducing Risks, Promoting Health Life* [38]. In its effort to help Member States raise healthy life expectancies, the report lays out an epidemiological case for increased governmental action on 10 leading causes of death worldwide, over half of which were related to NCDs. The centerpiece of the report is risk assessment, defined as “a systematic approach to estimating the burden of disease and injury due to different risks” [38]. The report details the specific methodology for characterizing risk and presents the burden of disease of 10 leading causes of disease burden worldwide. It establishes, through highly technical and specialized means, a method to calculate magnitude and origins of harm. Once harms can be known and characterized, they can be governed, as the report orders some specific interventions to manage risk factors based on their cost effectiveness. The report concludes that “the world is living dangerously” [38]. 

The risk rationality detailed in the 2002 World Health Report and in other documents are used to justify a specific form of governing through a program of actions and interventions which are decidedly illiberal. For example, the 2002 World Health Report [38] concluded that the risk assessment in its report:

“…offers a unique opportunity for governments. They can use it to take bold and determined actions against only a relatively few major risks to health, in the knowledge that the likely result within the next ten years will be large gains in healthy life expectancy for their citizens. Bold policies are required. They may, for example, have to focus on increased taxes on tobacco; legislation to reduce the proportion of salt and other unhealthy components in foods; stricter environmental controls and ambitious energy policies; and stronger health promotion and health safety campaigns” [38]. 

The illiberalism of NCD prevention and control extends beyond a greater regulatory role on the part of governments. The Global Action Plans of 2008 and 2013 [8,9] call for the greater integration of NCDs into all health and development planning, committing resources to addressing NCDs interventions, developing and strengthening national NCD programs, conducting assessment and evaluation, mobilizing non-health government agencies and strengthening multisectoral action, strengthening the health and social service workforce, creating accountability mechanisms, strengthening health systems, establishing a national NCD research agenda, and engaging in monitoring and surveillance of NCDs. The prescribed role of government in NCD prevention and control is expansive, as opposed to the liberal goal of government non-interference. Risk rationality is utilized to argue that behavioral risk factors are a result of social determinants across multiple social domains including but not limited to agriculture, education, employment, energy, environment, finance, foreign affairs social welfare, and trade and industry, making the argument that the dynamics of economy and society have led to increasing danger from NCDs. The premise that the social environment can be acted upon to reduce NCDs implies that economic and societal dynamics are not, in fact, natural but constructed and malleable, or that if they are natural, they do not lead to any social benefit. It is thus the charge of the state to become more expansive and to utilize scientific knowledge and an array of interventions to ensure the welfare of populations. The political rationality of the multilateral global NCD prevention and control movement is an assertion of a pastoral political rationality.

### 3.2. Mixing Governmentalities: Discipline, Police, and Neoliberalism in Policy Recommendations

Specific interventions were presented to Member States as “policy options” from which they could select based on national context, “taking into account region-specific situations and in accordance with national legislation and priorities and specific national circumstances” [9], though it is recognized that a comprehensive response would benefit all countries. The composition of the policy options to reduce risks from tobacco use, unhealthy diet, physical inactivity, and harmful alcohol use vary in the degree to which they draw upon governmentalities. Tobacco control policies draw heavily on police and other regulatory measures (e.g., smoke-free environments, taxation, advertising bans, contents regulation) combined with actions based on disciplinary government (e.g., warn people about dangers of smoking, offer cessation services). The policy mix to prevent harmful use of alcohol also relies on state regulatory powers (e.g., introducing and enforcing upper limit blood alcohol concentration, suspension of driving licenses, limiting availability of alcohol, regulating marketing of alcoholic beverages, and taxation) and disciplinary government (e.g., facilitating recognition of alcohol-related harm, public information and mass media campaigns, driver education and counseling). The policy mix to prevent unhealthy diet is a more varied mix of disciplinary measures (e.g., promoting breastfeeding, public campaigns and social marketing to inform and encourage healthier dietary practices, nutrition education, nutrition labeling), regulatory measures (e.g., limiting marketing of foods and non-alcoholic beverages to children, healthier composition of food, taxation), and neoliberal market-oriented interventions (e.g., increased availability of healthier food options). Efforts to promote physical activity rely heavily on disciplinary (e.g., physical education programs, campaigns to promote physical activity) and neoliberal orientations (e.g., creating environments more conducive to active transport and recreation). As global institutions reviewed the implementation of the broad set of recommendations, inadequate progress toward global goals became increasingly apparent [10,37] and greater emphasis was placed on a narrower set of highly cost effective “best buy” interventions for countries to adopt [43]. Similar to the 2013 Global Action Plan, the best buy interventions emphasized police power to reduce the harmful use of alcohol, police power supplemented by disciplinary measures for tobacco control, a mix of interventions to improve diet, and discipline (but not neoliberalism) to promote physical activity (see Table 2).

## 4. Discussion

Risk governmentality provides the justification for greater state intervention in economic and social domains. NCDs are characterized as largely being a function of four behavioral risks—tobacco use, unhealthy diet, physical inactivity, and harmful use of alcohol—and NCDs themselves a risk to health and well-being but also individual and state economic productivity and development [5]. Repeated across documents are statistics aimed at quantifying the magnitude and effect of risks across subpopulations, regions, and national income levels. The risks for NCDs are not only known, but they are governable and the measure for successful interventions merges health and economic goals through cost-effectiveness calculations. The recommended interventions are an amalgam of governmentalities, with the composition of political rationalities varying by risk factor. Interventions to reduce the harmful use of alcohol and tobacco use rely primarily on pastoral governmentality and less on discipline. Improving diet incorporates a mix of pastoral, disciplinary, and neoliberal governmentalities, and improving physical activity combines neoliberal and disciplinary governmentalities. 

The risk rationality employed to justify an expanded and interventionist state may not provide sufficient impetus for Member States to act accordingly. The WHO High-Level Commission on NCDs [7] has suggested that most obstacles to implementation are political, including “lack of political will, commitment, capacity, and action,” “lack of policies and plans,” “insufficient technical and operational capacity,” and “insufficient… financing to scale up national NCD responses” [7]. No amount of data on the burden of NCDs or evidence on cost effectiveness of recommended interventions may be able to overcome the political traditions or political economy of a country. In order to overcome the political barriers to adoption and implementation of recommended interventions, the evolution of the global NCD movement will need to expand from a focus on the technical dimensions of interventions to include the political determinants of health at the national and subnational level.

The call for greater attention to political barriers is hamstrung by the lack of empirical evidence on and funding for effective advocacy campaigns and increasing governing capacity. Some efforts at understanding political change for NCDs have been put forwards. For example, Reich [44] has advocated for the importance of normalizing a political economy of NCDs approach. He suggests that a political economy approach can be helpful for understanding the commercial determinants of health, promoting social movements, and setting national priorities and governing government agencies. Applying Shiffman and Smith’s [45] framework for generating political priority for global health initiatives, Maher and Sridhar [46] emphasize strategic communication as a core activity of global health policy communities. They argue for NCD issue portrayals that uniquely target political leaders through a vast array of channels. In their application of Shiffman and Smith [45], Heller and colleagues [47] suggest that the economic argument has more traction with the broader policy community than health concerns but that success will require fostering civil society and developing a broader and more inclusive global governance structure. 

The present analysis adds to this literature by suggesting that characteristics of policies themselves—their implied governmentality—influence their political acceptability. Taken as a whole, the order of discourse which constitutes global NCD policy recommendations would require governmental openness to a variety of political rationalities and develop different technologies of government to fully implement a comprehensive NCD agenda (sans a liberal governmentality). In emphasizing that national priorities and circumstances will determine which policy options Member States adopt in the 2013 Global Action Plan, the onus was placed on domestic actors to do the work of softening political and governmental systems to an array of governmentalities. In contrast to the critical approach of CDA which argues that the ideologies compete with one another, a comprehensive NCD agenda would require varying ideologies to exist complementarily. A constant framing of policy options as a suite of complementary and interdependent efforts and not as discrete and independent policies will be needed to achieve this. A commitment of resources from governments, development agencies and other funders to support advocacy efforts and strengthen and expand governmental infrastructure—especially the recruitment and training of public health workers—can ensure that policy adoption is translated into effective governing. Technical assistance and sharing of best practices for developing and sustaining technologies of government to support multiple governmentalities will also be needed to integrate multiple governmentalities into governments’ repertoire of capabilities. An emphasis on multiple governmentalities in addressing political barriers to reducing NCD burden is a strategy that adds the long-term goal of having a government responsive to health interests in a variety of forms to the short-term goals of specific policy adoption.

Neoliberalism as a global economic system has been a point of focus for critics who argue it is a major driving force of NCDs [48,49,50]. Springer [51] notes, however, that the term “neoliberalism” is used in numerous ways including as an ideological hegemonic project, policy and program, state form, or as governmentality. Neoliberalism as governmentality used in this analysis should not be confused with neoliberalism as hegemonic ideological project or policy and program critiqued by others. Foucault [30] asserted, however, that neoliberal governmentality was extending beyond the bounds of the economy and becoming a dominant form of governmentality. The concept of multiple governmentalities resists tendencies to rely on a single governmentality to address NCDs. Reliance on neoliberal governmentality would result in corporate interests that would use their financial and market power to overwhelm governmental efforts to create healthier choice environments. Dependence on discipline would lead to a form of prudentialism which shifts responsibility for minimizing risk to individuals; freedom becomes equated with personal responsibility [32], while ignoring the powerful commercial determinants of NCDs. However, the role of individual agency and motivation is overlooked when relying on pastoral power. Working toward a multiple governmentalities approach promotes and facilitates a balanced, comprehensive NCD agenda. 

We have argued elsewhere of the importance of fundamental structural reforms to reduce the burden of NCDs, including those which allow for alternative understandings of political challenges and more shared power [5]. Structures of power need to be meaningfully opened to traditionally excluded and underrepresented groups as a means of introducing and legitimizing alternative forms of productive power [52]; the involvement of civil society groups in the HLM reflects a step in that direction. Politically dominant groups will have to be convinced that opening power structures to new modes of governance can inform improved governance and have whole-of-society benefits. More inclusive decision making will play a significant role in convincing Member States that multiple governmentalities need not be in competition with one another to be the singular approach to reducing the burden of NCDs. The political efforts of a broad group of stakeholders will be needed to develop policy innovations in which multiple governmentalities within an order of discourse can be complementary and synthesized in ways which productively address NCDs.

This study was limited to official documents from a global fora of nation states. While the documents were deliberative consensus statements, we were unable to detail the negotiation of the documents and the ideological competition from specific actors because negotiations often take place in private and records of such discussions are never made public. Thus, we are unable to link competing ideologies to specific Member States or other actors to assess how ideologies are linked to power relationships in the negotiation of reviewed documents. We are also unable to ascertain the degree to which final recommendations are truly reflective of a belief in their empirical foundations compared to the political interests of Member States to include or exclude particular recommendations. The findings from this analysis suggest that comprehensive efforts to reduce the burden of NCDs will require the adoption of multiple governmentalities. This approach to facilitating policy adoption and effective implementation needs empirical testing through detailed cases studies of national contexts as well as aggregating country-level policy adoption data such as the NCD Country Profiles and analyzing them with respect to domestic political, governmental, and civil society characteristics. 

## 5. Conclusions

The rapid development of the global policy infrastructure to address NCDs has not resulted in similar achievements at national and subnational levels to take tangible action on combating these diseases of growing global health burden. Thus, progress toward stemming the tide of NCDs has been inadequate, and commitments to prevent and control NCDs are insufficient to meet target 3.4 of the Sustainable Development Goals [10], which establishes a broader international consensus around the need to address NCDs within global goals to promote sustainability. Even with a strong empirical base of known and effective interventions, attention must focus on how to translate recommended interventions into policy adoption and effective implementation. A first step will be to increase acceptability for the political rationalities embodied in policy options through increased attention, increasing public acceptability, and softening the political infrastructure to various approaches. Policy adoption must be followed with developing the necessary norms, procedures, and institutions appropriate to the political rationalities of specific interventions and measuring effectiveness and compliance to NCD goals. Concurrent use of technologies of the government based on different governmentalities will require flexible, multidimensional governmental systems. The support of Member States will, therefore, need to move beyond technical interventions and diversifying and strengthening the political aptitude and resourcefulness of actors working to extend progress toward reducing NCDs into national and subnational contexts. 

## Figures and Tables

**Table 1 ijerph-17-04413-t001:** List of study documents.

Year	Author	Title	Type
1997	WHO	The World Health Report: Conquering Suffering, Enriching Humanity	Descriptive
1998	WHA	WHA51.18–Noncommunicable disease prevention and control	Political
2000	WHA	WHA53.17–Prevention and control of noncommunicable diseases	Political
2000	WHO	Global strategy for the prevention and control of noncommunicable diseases	Prescriptive
2002	WHO	The World Health Report 2002: Reducing Risks, Promoting Healthy Life	Descriptive
2003	WHO/ UNICEF	Global Strategy for Infant and Young Child Feeding	Prescriptive
2003	WHO	WHO Framework Convention on Tobacco Control (FCTC)	Prescriptive
2004	WHO	Global Strategy on Diet, Physical Activity, and Health	Prescriptive
2005	WHO/ Public Health Agency of Canada	Preventing Chronic Diseases. A vital investment	Prescriptive
2007	WHO	FCTC Global Progress Report 2007	Descriptive
2008	WHO	2008–2013 Action Plan for the Global Strategy for the Prevention and Control of Noncommunicable Diseases	Prescriptive
2008	WHO	WHO Report on the Global Tobacco Epidemic, 2008: The MPOWER Package	Prescriptive
2008	WHO	FCTC Global Progress Report 2008	Descriptive
2009	WHO	WHO Report on the Global Tobacco Epidemic, 2009: Implementing Smoke-free Environments	Descriptive
2009	WHO	FCTC Global Progress Report 2009	Descriptive
2010	WHO	Global Status Report on Noncommunicable Diseases 2010	Descriptive
2010	WHO	Global Strategy to Reduce the Harmful Use of Alcohol	Prescriptive
2010	WHO	Set of Recommendations on Marketing of Foods and Non-Alcoholic Beverages to Children	Prescriptive
2010	WHO	FCTC Global Progress Report 2010	Descriptive
2011	WHO	WHO Report on the Global Tobacco Epidemic, 2011: Warning About the Dangers of Tobacco	Descriptive
2011	WHO/ Ministry of Public Health and Social Development of Russian Federation	Moscow Declaration	Political
2011	UN	Political Declaration of the High-Level Meeting of the General Assembly on the Prevention and Control of Non-communicable Diseases.	Political
2011	WHO	Scaling Up Action Against Noncommunicable Disease: How Much Will It Cost?	Prescriptive
2012	WHO	FCTC Global Progress Report 2012	Descriptive
2013	WHO	Global Action Plan for the Prevention and Control of Noncommunicable Diseases 2013–2020	Prescriptive
2013	WHO	WHO Report on the Global Tobacco Epidemic, 2013: Enforcing Bans on Tobacco Advertising, Promotion and Sponsorship	Descriptive
2014	WHO	Noncommunicable Diseases Country Profiles 2014	Descriptive
2014	WHO	Global Status Reports on Noncommunicable Diseases	Descriptive
2014	WHO	FCTC Global Progress Report 2014	Descriptive
2014	UN	Outcome Document of the High-Level Meeting of the General Assembly on the Reviewe of the Progress Achieved in the Prevention and Control of Non-Communicable Diseases	Political
2014	WHO	EB136/8–Outcome of the Second International Conference on Nutrition	Political
2015	WHO	WHO Report on the Global Tobacco Epidemic, 2015: Raising Taxes on Tobacco	Descriptive
2016	WHO	Global Report on Diabetes	Descriptive
2016	WHO	FCTC Global Progress Report 2016	Descriptive
2016	UN	A/RES/70/259–United Nations Decade of Action on Nutrition (2016–2025)	Political
2016	Global Conference on Health Promotion	Shanghai Declaration on Promoting Health in the 2030 Agenda for Sustainable Development	Political
2017	Presidencia Republica Oriental del Uruguay/ WHO	Montevideo Roadmap 2018–2030 on NCDs as a Sustainable Development Priority	Political
2017	WHO	A70/31–Report of the Commission on Ending Childhood Obesity: implementation plan	Political
2017	WHO	Tackling NCDs–‘Best Buys’ and Other Recommended Interventions for the Prevention and Control of Noncommunicable Diseases	Prescriptive
2017	WHO	WHO Report on the Global Tobacco Epidemic, 2017: Monitoring Tobacco Use and Prevention Policies	Descriptive
2017	WHO	Noncommunicable Disease Progress Monitor 2017	Descriptive
2018	WHO	Noncommunicable Disease Country Profiles 2018	Descriptive
2018	WHO	FCTC Global Progress Report 2018	Descriptive
2018	UN	A/RES/73/2–Political declaration of the third high-level meeting of the General Assembly on the prevention and control of non-communicable diseases	Political
2018	WHO	Global Action Plan on Physical Activity 2018-2030	Prescriptive
2018	WHO	Saving Lives, Spending Less: A Strategic Response to Noncommunicable Diseases	Prescriptive
2018	WHO Independent High-Level Commission on Noncommunicable Disease	Time to Deliver. Report of the WHO Independent High-Level Commission on Noncommunicable Disease	Prescriptive
2018	WHO/ WHO Independent High-Level Commission on Noncommunicable Disease	19 Bold Recommendations for Heads of State and Government to Accelerate Action on Reaching Target 3.4 on NCDs by 2030	Prescriptive
2018	WHO/ WHO Independent High-Level Commission on Noncommunicable Disease	Think Piece: Why is 2018 a Strategically Important Year for NCDs?	Prescriptive

WHA: World Health Assembly; WHO: World Health Organization; UN: United Nations; UNICEF: United Nations Children’s Fund.

**Table 2 ijerph-17-04413-t002:** WHO “Best buys” and other recommended interventions, WHO, 2018, by governmentality type.

GovernmentalityType	Reduce Tobacco Use	Reduce Harmful Use of Alcohol	Reduce Unhealthy Diet	Reduce Physical Inactivity
Discipline	Implement effective mass media campaigns that educate the public about the harms of smoking/tobacco use and secondhand smokeProvide cost-covered, effective and population-wide support (including brief advice, national toll-free quit line services) for tobacco cessation to all those who want to quit	Provide brief psychosocial intervention for persons with hazardous and harmful alcohol use	Reduce salt intake through a behaviour change communication and mass media campaignReduce salt intake through the implementation of front-of-pack labelling	Implement community wide public education and awareness campaign for physical activity which includes a mass media campaign combined with other community-based education, motivational and environmental programmes aimed at supporting behavioural change of physical activity levelsProvide physical activity counselling and referral as part of routine primary health care services through the use of a brief intervention
Pastoral/Police	Increase excise taxes and prices on tobacco productsImplement plain/standardized packaging and/or large graphic health warnings on all tobacco packagesEnact and enforce comprehensive bans on tobacco advertising, promotion and sponsorshipEliminate exposure to second-hand tobacco smoke in all indoor workplaces, public places, public transport	Increase excise taxes on alcoholic beveragesEnact and enforce bans or comprehensive restrictions on exposure to alcohol advertising (across multiple types of media)Enact and enforce restrictions on the physical availability of retailed alcohol (via reduced hours of sale)Enact and enforce drink-driving laws and blood alcohol concentration limits via sobriety checkpoints	Reduce sugar consumption through effective taxation on sugar-sweetened beveragesEliminate industrial trans-fats through the development of legislation to ban their use in the food chainReduce salt intake through the reformulation of food products to contain less salt and the setting of target levels for the amount of salt in foods and meals	
Neoliberalism			Reduce salt intake through the establishment of a supportive environment in public institutions such as hospitals, schools, workplaces and nursing homes, to enable lower sodium options to be provided

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
