# Peer review of "Governing Noncommunicable Diseases Through Political Rationality and Technologies of Government: A Discourse Analysis"

_ijerph, 2020, doi:10.3390/ijerph17124413_

Round 1
Reviewer 1 Report
Originality
In my opinion the research question is ill-defined in this research. It is not clear to me what the research problem is not how the paper aims to address that problem. Notwithstanding this criticism, discourse analysis is often merely descriptive and this might need to be highlighted at the beginning of the research. Ideally there would be some statement of problem that is being solved by this research (i.e. a gap in the body of knowledge that needs to be filled). As a result of what is fundamentally descriptive research, I am not convinced that the results provide an advance on current knowledge. This is particularly because there’s only one organisation principally outlined in the “discourse analysis” and the discourse has at least the elements of a conversation in theoretical terms. What this means is I would expect to see more discussion about particular protagonists and their positions within the discourse. Fundamentally this paper presents a descriptive analysis of World Health Organisation and United Nations documents. In my opinion this does not constitute a discourse.
Significance
The significance of the research is not well explained in the paper itself. That said the conclusions are justified and supported by the results there are no hypotheses and speculations to identify. This is appropriate for a paper of this type.
Quality of presentation
The overall quality of presentation is acceptable.
Scientific soundness
As mentioned previously, I am not convinced about the scientific soundness or the range of discovery undertaken in this paper. The methodology was appropriate for a discourse analysis. However in my opinion the results are under reported.
Interest to the readership
The paper could be of interest to the readership of the International Journal of environmental research and public health. However, some more persuasive discussion about the contributions of this paper to extant practice or knowledge would be helpful. I am left guessing why I need to read this paper and what its importance is in terms of research or practice of public health or environmental research. In my opinion, is too much description of theories and not enough application of those theories to public policy (or public health). I suspect that the paper will only be of interest to a limited number of people and will not be widely read.
Overall merit
The paper is reasonably well written and with some work (as outlined previously) will have merit. The researchers have not addressed an important long-standing question with this research.
English level
The English level is acceptable. The paper was easy to read.
Reviewer 2 Report
Congratulations for this very interesting article
Reviewer 3 Report
Thank you for the opportunity to review this interesting and well-written manuscript. This article is rigorous and provides unique evidence in this space. However, there are a couple of areas for improvement for consideration, including:
1) The introduction section provides a detailed background of the international policy context and relevant theory, however, there were several sentences that require minor re-wording:
L50-Remove ‘and’ before "and policy adoption"
L72 This sentence is difficult to read, perhaps a word is missing? I suggest rephrasing
L76 Should "bring" be "bringing"? Otherwise rephrase
L88 Foucault quote seems incomplete. Population what? Or is it [the] population?
L173 "operate" should be "operates"
2) The introduction would also be enhanced if it included a discussion of whether and (how) governmentality has previously been used in analyses of international agencies/policy? Or for analysis of nation-level governance of public health policy?
3) Similarly, have governmentality types previously been used to categorise NCD prevention policy options? Examples of NCD policy action by governmentality type would be useful in introduction after each governmentally type is defined.
4) The beginning of the methods section needs a definition of CDA as the analytical method for examining govermentatility
5) Was the analysis purely deductive (if so this needs to be made clear). Alternatively were you open to any new codes related to discourse/governmentality being identified from the data?
6) Examples of neoliberal Physical Activity policy options are missing from Results table 2 (but are mentioned in the manuscript text)
7) The Discussion argues that much of the discourse/governmentality is at odds with the liberalism that is dominant in many nation states. If this is the case, why has uptake of tobacco control policies been reasonably popular even though these use police/pastoral approach? Some discussion of the differing uptake of policy options (such as tobacco vs diet) and whether this reflects governmentality would strengthen the discussion section.
8) Line 399-400 seems incomplete: “The evolution of the global NCD movement will need to shift from the technical to the political determinants of health at the national and subnational level.” Why do they need to make this shift?
9) The discussion section would also be strengthened by greater engagement with other NCD Prevention policy analysis literature. The work of Phillip Baker, Ann Marie Thow, Jeremy Shiffman and Michael Reich for example.
10 The conclusion would be strengthened by clearer implications for policy advocacy and future research. For example, how should advocates frame arguments in order to generate political support for NCD policy adoption?
